# The role of exosomal miRNA-125b derived from colon cancer-associated fibroblasts in skeletal muscle cachexia

Ho Seung Kim[1], Jinsu Kim[1], Bom Lee[2], Yonghyun Lee[2], So-Yeon Park[2], Bo-Young Oh[3], Kyung-Ah Cho[4], Soon Sup Chung[1], Gyoung Tae Noh[1]*

1 Department of Surgery, Ewha Womans University College of Medicine, Seoul, South Korea,
2 Department of Pharmacy, College of Pharmacy, Ewha Womans University, Seoul, South Korea,
3 Department of Surgery, Hallym University College of Medicine, Seoul, South Korea, 4 Department of Microbiology, Ewha Womans University College of Medicine, Seoul, South Korea

* nogang@ewha.ac.kr

## Abstract

Cancer-associated cachexia is a multifactorial syndrome characterized by significant weight loss, primarily due to skeletal muscle atrophy. This condition impairs the quality of life and survival of patients with cancer. Although the mechanisms underlying cancer-associated cachexia, including exosomes and microRNAs (miRNAs), have been extensively explored, research specifically focusing on cancer-associated fibroblast (CAF)-derived exosomes is lacking. Therefore, in this study, we evaluated the effects of CAF-derived exosomal miRNAs from colon cancer on skeletal muscles using the Human Skeletal Muscle (HSkM) cell line. CAF-derived exosomes were isolated from colon cancer samples, and their effects on cell morphology were analyzed using confocal microscopy. The results indicate that treatment with CAF-derived exosomes significantly reduced myosin diameter. Moreover, miRNA sequencing revealed that miR-125b was enriched in CAF-derived exosomes. HSkM cells were subsequently transfected with a miR-125b mimic, which significantly reduced myosin diameter. Notably, co-treatment with CAF-derived exosomes and an miR-125b inhibitor reversed this effect. In conclusion, this study demonstrates the potential role of CAF-derived exosomes and miR-125b in cancer-associated cachexia, offering insights into the contribution of the tumor microenvironment and suggesting possible therapeutic targets.

## Introduction

Cancer-associated cachexia is a multifactorial and often irreversible syndrome characterized by significant weight loss, primarily driven by skeletal muscle atrophy [1,2]. This condition diminishes quality of life and adversely affects overall survival in patients [3,4]. Cachexia is induced through a complex pathway [5], with

**Data availability statement:** All relevant data are within the paper and its Supporting Information file.

**Funding:** National Research Foundation of Korea (NRF) grant funded by the Korea government (MSIT) (2022R1G1A1011832) and Ewha Womans University Research Grant 2022. The funders had no role in study design, data collection and analysis, decision to publish, or preparation of the manuscript.

**Competing interests:** The authors have declared that no competing interests exist.

several factors and mechanisms potentially involved in vitro, including tumor-secreted factors [6], pro-inflammatory cytokines [7], hormonal dysregulation [8], anticancer chemotherapeutic agents [9], and microRNA (miRNA)-mediated gene regulation [10].

Recently, the role of miRNAs, which are small non-coding RNAs that regulate gene expression via translational repression or messenger RNA (mRNA) degradation [11], in the development of cachexia has been highlighted. miRNAs are involved in various cancer-related processes, including proliferation [12], apoptosis [13], migration, and invasion [14]. Notably, miRNAs play a role in cachectic muscle wasting [15,16]. In particular, miRNAs secreted within exosomes regulate gene expression in recipient cells by targeting their mRNAs [17]. These exosomes, a subset of tumor-secreted extracellular vesicles measuring 30–150 nm in diameter, mediate intercellular communication by encapsulating proteins, lipids, metabolites, and miRNAs [17–19].

The tumor microenvironment (TME) consists of various stromal and immune cell types that influence tumor behavior and systemic physiology. In cancer cachexia, these cells, including endothelial cells, immune cells, adipocytes, and cancer-associated fibroblasts (CAFs) in particular, secrete inflammatory cytokines and remodeling factors that can induce systemic metabolic disturbances [20,21]. CAFs are the most abundant stromal cells in the TME and produce interleukin (IL)-6, IL-8, tumor necrosis factors (TNF)-α, and C-C motif chemokine ligand 2, all of which contribute to chronic inflammation and tumor progression [20]. CAFs actively communicate with other cells and can reshape the TME [22]. Exosomes released from these fibroblasts increase the invasive behavior of cancer cells [23].

Despite their abundance and activity, CAF-derived exosomes have rarely been studied in the context of cachexia. Although several studies have implicated tumor cell-derived or circulating exosomes in the development of cancer-associated cachexia, the specific contribution of exosomes released by CAFs remains largely unexplored. In particular, whether CAF-derived exosomal miRNAs directly influence skeletal muscle atrophy has not been clarified. Therefore, in this study, we aimed to investigate the effects of CAF-derived exosomes from colon cancer tissues on skeletal muscle cells, with a particular focus on exosomal miRNAs.

## Materials and methods

### Patients and samples

All samples and information were collected from patients who underwent surgery for colon cancer between 17/06/2022 and 16/06/2023 and had histologically confirmed adenocarcinomas. Patients who underwent preoperative treatment, such as chemotherapy, radiotherapy, or endoscopic resection, were excluded. The study protocol for the experimental use of patient samples and information was approved by the Institutional Review Board of the Ewha Womans University Seoul Hospital [SEUMC 2022-04-028]. Samples from patients were only used experimentally after obtaining written informed consent.

## Isolation of CAFs

CAFs were isolated from tumor samples obtained from patients. Tumor biopsies were washed with phosphate-buffered saline (PBS) containing 10% penicillin/streptomycin, minced into approximately 1 mm³ pieces, and dissociated into single-cell suspensions using a gentleMACS Octo Dissociator (130-096-427; Miltenyi Biotec, Bergisch Gladbach, Germany) in combination with a human tumor dissociation kit (130-095-929; Miltenyi Biotec) at 37°C for 1 h. The dissociated tissue was filtered through a 70-μm strainer, centrifuged at 300 × g for 7 min, and resuspended in MACS microbead buffer. Fibroblasts were isolated using anti-fibroblast microbeads (130-050-601; Miltenyi Biotec), a MACS Separator, and MS Columns (130-042-201; Miltenyi Biotec). Isolated cells were cultured in RPMI 1640 medium (LM 011−01; Welgene, Daegu, Republic of Korea) supplemented with 10% fetal bovine serum (FBS; 16000–044; Gibco, Grand Island, NY, USA) and 1% penicillin/streptomycin (LS 202−02; Welgene) on poly-D-lysine-coated dishes (22100; SPL Life Sciences, Republic of Korea) at 37°C in a 5% $CO_2$ incubator. CAFs from passages 1–10 were used for experiments.

## Immunoblotting

Total CAF cell lysates and exosomes were prepared using RIPA buffer (Cell Signaling Technology [CST], Danvers, MA, USA), and protein concentrations were quantified using the Pierce™ BCA Protein Assay Kit (23225; Thermo Fisher Scientific, Waltham, MA, USA). Equal protein amounts were separated on 10% SDS-PAGE gels, transferred to PVDF membranes (Cytiva, Marlborough, MA, USA), and probed with antibodies for alpha smooth actin (α-SMA; D4K9N, 19254S; CST) and beta-actin (4970S; CST), which are CAF markers; and CD9 (EPR2949, ab92726; Abcam, Cambridge, UK) and CD63 (MX-49.129.5, ab193349; Abcam), which are exosomal markers. After washing with 1 × Tris-buffered saline with Tween 20 (TR2007-105-80; Biosesang, Gyeonggi, Republic of Korea), the membranes were incubated with horseradish peroxidase-conjugated secondary antibodies (CST). Proteins were detected using SuperSignal™ West Atto Substrate (a38554; Thermo Fisher Scientific) and visualized using a Bio-Rad ChemiDoc Imaging System (Bio-Rad Laboratories, Hercules, CA, USA).

## Flow cytometry

CAFs were labeled with an anti-fibroblast activation protein (FAP) antibody (Mouse monoclonal IgG1, 427819; Novus Biologicals, Littleton, CO, USA) and subsequently stained with a fluorescein isothiocyanate-conjugated anti-mouse IgG1 secondary antibody. After staining, the cells were centrifuged at 400 × g for 5 min at room temperature, fixed with 1% paraformaldehyde in flow cytometry staining buffer (0.5% FBS in PBS), and analyzed using a NovoCyte flow cytometer (ACEA Biosciences, San Diego, CA, USA) with NovoExpress software.

## Exosome purification

Cells were cultured in RPMI 1640 medium (LM 011–01; Welgene) supplemented with 10% exosome-depleted FBS, prepared through ultracentrifugation of standard FBS (16000–044; Gibco) at 120,000 × g and 4°C for 18 h using an XE-90 ultracentrifuge (Beckman Coulter, Brea, CA, USA) with a Type 41 Ti swing rotor. The culture supernatants were collected and subjected to ultracentrifugation at 120,000 × g and 4°C for 120 min to isolate exosomes. The exosome pellet was resuspended in PBS and stored at −80°C until further use. The exosome concentration was quantified using a Pierce BCA Protein Assay Kit (23225; Thermo Fisher Scientific).

## Transmission electron microscopy

CAF exosomes were identified using transmission electron microscopy (TEM). Purified exosomes were diluted in PBS at a 1:1000 ratio, and 5 μL of the diluted sample was placed onto Formvar-carbon-coated electron microscopy grids. The grids were stained with 2% uranyl acetate, and excess stain was removed using filter paper. The morphology and size

of exosomes were visualized using an H-7650 transmission electron microscope (Hitachi, Tokyo, Japan) at 80 kV. Digital images with scale bars were captured at magnifications of 70,000–200,000 ×.

## Nanoparticle tracking analysis

Three samples were diluted 10 × with distilled water and filtered using a 0.45-µm syringe filter. The size and zeta potential of the exosomes in the samples were measured using a Zetasizer Nano ZS90 (v7.13, Malvern Instruments Ltd., Malvern, UK). Controls for the corresponding samples were measured using the same method.

## Cell viability assay

For cell viability assays, $2 \times 10^3$ CAFs or Normal Human Skeletal Myoblasts (HSkM, A12555; Gibco) were seeded into 96-well plates (030096; SPL Life Sciences) and incubated overnight at 37°C with 5% $CO_2$. The next day, the cells were treated with CAF-derived exosomes or transfected with miRNA in fresh media for 48 h. On the indicated day, 10 µL of EZ-CYTOX reagent (EZ-1000; DoGenbio, Republic of Korea) was added to each well and incubated for 1 h. Absorbance was then measured at 450 nm using an ELISA plate reader (SpectraMax ABS Plus, San Jose, CA, USA).

## Confocal microscopy

HSkM cells ($2 \times 10^4$) were seeded onto four-well chamber slides and incubated overnight at 37°C with 5% $CO_2$. After washing with PBS, cells were treated with 25 µg/mL CAF-derived exosomes or subjected to miRNA transfection for 48 h. Cells were fixed with 4% paraformaldehyde, permeabilized with 0.1% Triton X-100 (T8787; Sigma-Aldrich, St. Louis, MO, USA), blocked with 2% bovine serum albumin, and stained with Alexa Fluor 488-conjugated Myosin 4 antibody (53-6503-82; Invitrogen, Carlsbad, CA, USA) at a 1:250 dilution overnight at 4°C. The myotubes were mounted with Fluoroshield containing 4',6-diamidino-2-phenylindole (DAPI; F6057, Sigma-Aldrich, Germany), imaged using a Zeiss LSM 800 confocal microscope (Zeiss, Oberkochen, Germany), and analyzed using the ZEN software. Average myotube diameters from at least 15 random fields per well were measured using the ImageJ software (National Institutes of Health, Bethesda, MD, USA).

## Next-generation sequencing of exosomal miRNAs from CAFs

RNA from CAF-derived exosomes was used to construct small RNA libraries using the SMARTer smRNA-Seq Kit for Illumina (635030; Takara Bio, Shiga, Japan) according to the manufacturer's instructions. Libraries were prepared through polyadenylation, complementary DNA synthesis, and polymerase chain reaction amplification and validated for size, purity, and concentration using an Agilent Bioanalyzer 2100 (Agilent Technologies, Santa Clara, CA, USA). Equimolar amounts of libraries were pooled and sequenced using Illumina NovaSeq6000 (Illumina Inc., San Diego, CA, USA). Image decomposition and quality scoring were performed using the Illumina pipeline. Known miRNAs were identified using miRBase v22.1, and other RNA sequences were classified using RNAcentral 14.0.

## Exosome treatment and transfection of miRNA mimic and inhibitor in HSkM cells

HSkM cells were transfected with an hsa-miR-125b-5p mimic (20 nM), hsa-miR-125b-5p inhibitors (50 nM), or mock controls using Lipofectamine RNAiMAX (13778075; Thermo Fisher Scientific), according to the manufacturer's protocol. The mimic or inhibitor was mixed with Lipofectamine RNAiMAX in Opti-MEM (31985062; Thermo Fisher Scientific) prior to transfection. The sequence of the hsa-miR-125b-5p mimic (GenePharma, Shanghai, China) was 5′-UCCCUGAGA CCCUAACUUGUGA-3′, and the sequence of the hsa-miR-125b-5p inhibitor was 5′-UCACAAGUUAGGGUCUCAGGGA-3′.

## Statistical analysis

Statistical analyses were performed using the GraphPad Prism 5.0 software (GraphPad Inc., La Jolla, CA, USA). Data are expressed as mean ± standard deviation (SD) or standard error of the mean (SEM). Statistical significance was

determined using one-way analysis of variance followed by Tukey's post hoc test, with $p < 0.05$ considered statistically significant (* $P < 0.05$,** $P < 0.01$, and *** $< 0.001$).

## Results

### Isolation and validation of CAFs

We successfully isolated spindle-shaped fibroblasts from five patients with colon cancer (Fig 1A, 1B). The isolated fibroblasts were positive for α-SMA and β-actin, indicating consistency with the characteristics of CAFs (Fig 1C). Flow cytometry successfully detected the expression of FAP in CAFs, further confirming the identity of the CAFs (Fig 1D).

### Isolation and validation of CAF-derived exosomes

We isolated and purified exosomes from CAFs. We observed vesicles smaller than 200 nm, which is consistent with the typical size range of exosomes observed using microscopy (Fig 2A). We further confirmed the expression of the exosomal

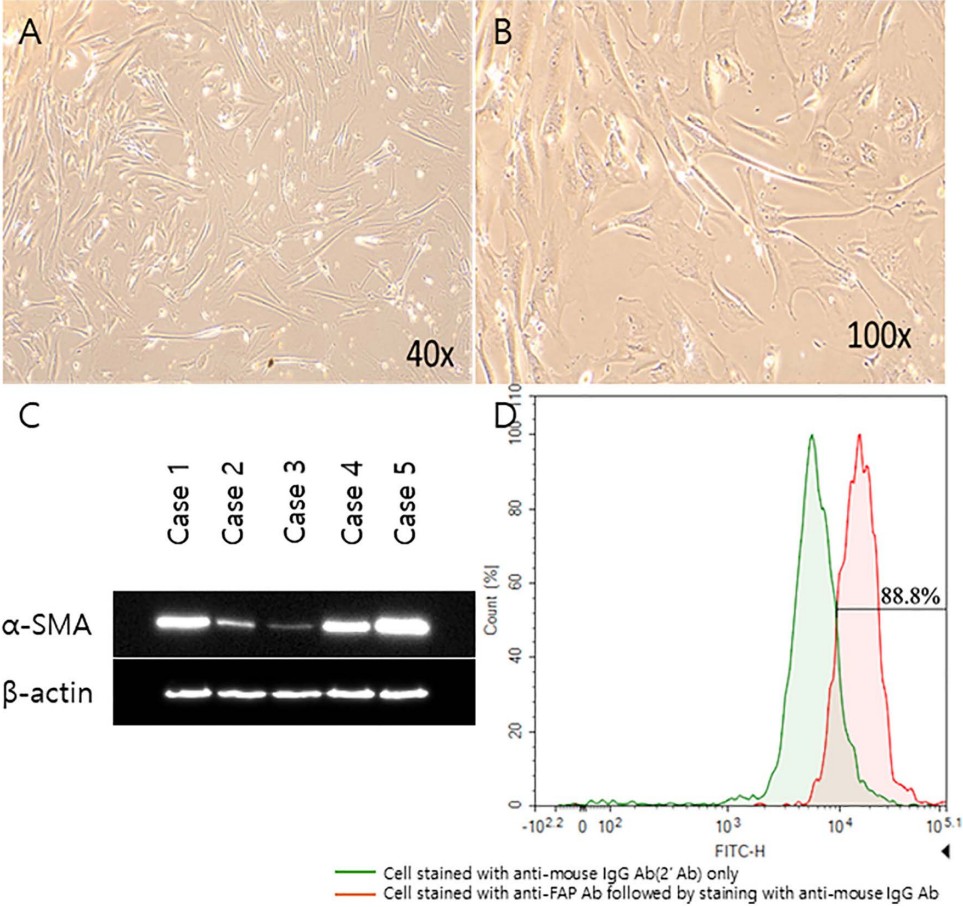

**Fig 1. Isolation and validation of CAFs from colon cancer samples. (A)** Microscopic images of CAFs at 40 × **(A)** and 100 × **(B)** magnifications. **(C)** Western blot analysis of the expression of α-SMA (42 kDa), a CAF marker; β-actin (45 kDa) was used as a loading control. **(D)** Flow cytometry analysis of FAP expression in isolated CAFs. The red peak indicates cells stained with anti-FAP antibody, and the green peak represents cells stained with isotype control (anti-mouse IgG Ab). CAFs, cancer-associated fibroblasts; α-SMA, alpha smooth actin; FAP, fibroblast activation protein.

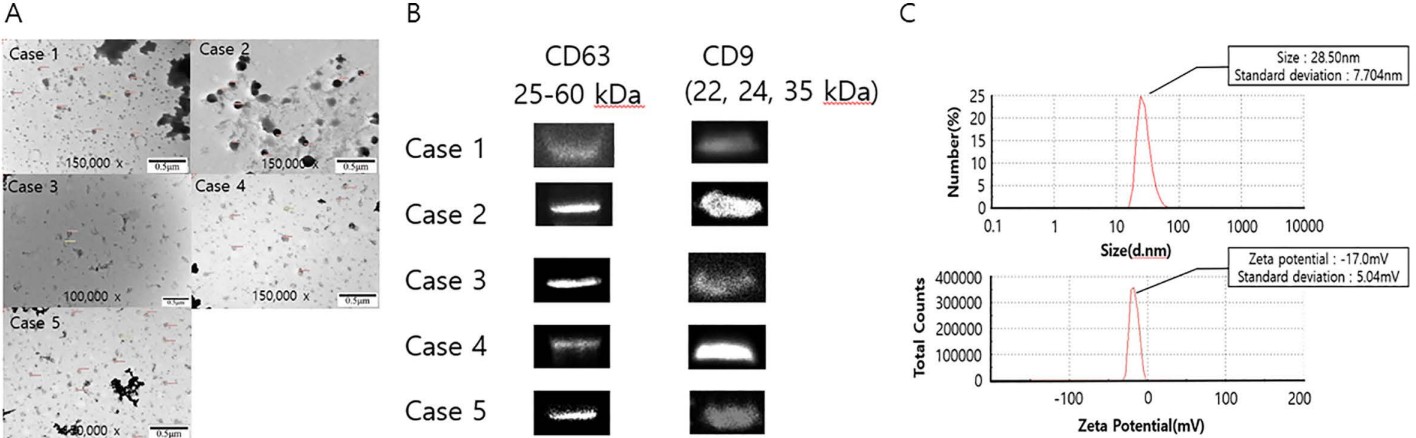

**Fig 2. Isolation and validation of exosomes from CAFs. (A)** Transmission electron microscopy image showing the characteristic cup-shaped morphology of CAF-derived exosomes (scale bar = 0.5 μm). **(B)** Immunoblotting assays of the expression of the exosome markers CD63 (25–60 kDa), and CD9 (22, 24, 35 kDa). **(C)** Nanoparticle tracking analysis illustrates the size distribution (mean size: 28.50 nm, standard deviation: 7.70 nm) and zeta potential (mean zeta potential: −17.0 mV, standard deviation: 5.04 mV) of the isolated exosomes. This analysis was conducted approximately 6 months after initial exosome extraction. Delayed processing may have contributed to the smaller mean particle size observed. CAFs, cancer-associated fibroblasts.

surface markers CD9 and CD63 using immunoblotting (Fig 2B). Nanoparticle tracking analysis further validated the particle size distribution and concentration of the isolated exosomes (Fig 2C).

**Treating HSkM cells with CAF-derived exosomes**

We assessed cell viability to determine the optimal exosome concentration that does not induce significant cell death. Specifically, we assessed viability 48 h after exosome treatment, with each condition tested independently at least three times in patient-derived cells. Treatment with 25 μg/mL and 50 μg/mL exosomes resulted in mean cell viabilities of 79.5% and 63.3%, respectively (Fig 3). In all five patient-derived cases, as well as in the total pooled analysis, treatment with 50 μg/mL exosomes resulted in a significant decrease in cell viability, whereas treatment with 25 μg/mL exosomes maintained cell viability while still demonstrating biological effects.

We then treated HSkM cells with CAF-derived exosomes. Confocal microscopy revealed that control cells displayed normal myosin structure, whereas cells treated with 25 μg/mL and 50 μg/mL exosomes exhibited reduced myosin diameters (Fig 4A), which were significantly smaller than those in the control group (Fig 4B). For subsequent experiments involving co-treatment with exosomes and miRNA inhibitors, we selected 25 μg/mL as the optimal exosome concentration as it represents the minimum effective concentration that produced a measurable effect while maintaining high cell viability. This concentration was selected to minimize potential cytotoxic effects and ensure the specificity of the observed outcomes.

**miRNA analysis**

We performed miRNA sequencing of CAF-derived exosomes from a single sample. High-sensitivity analysis of DNA fragments confirmed the quality and quantity of the gel-recovered library (Fig 5A). Fig 5B depicts the small RNA (smRNA) composition of the sample, revealing the relative proportions of smRNA classes, including known miRNA, candidate miRNA, ribosomal RNA, transfer RNA, small nuclear RNA, and small nucleolar RNA, classified based on the processed reads. In total, 370 (0.19%) known miRNAs were identified (Fig 5C). The miRNAs with the highest read counts in the

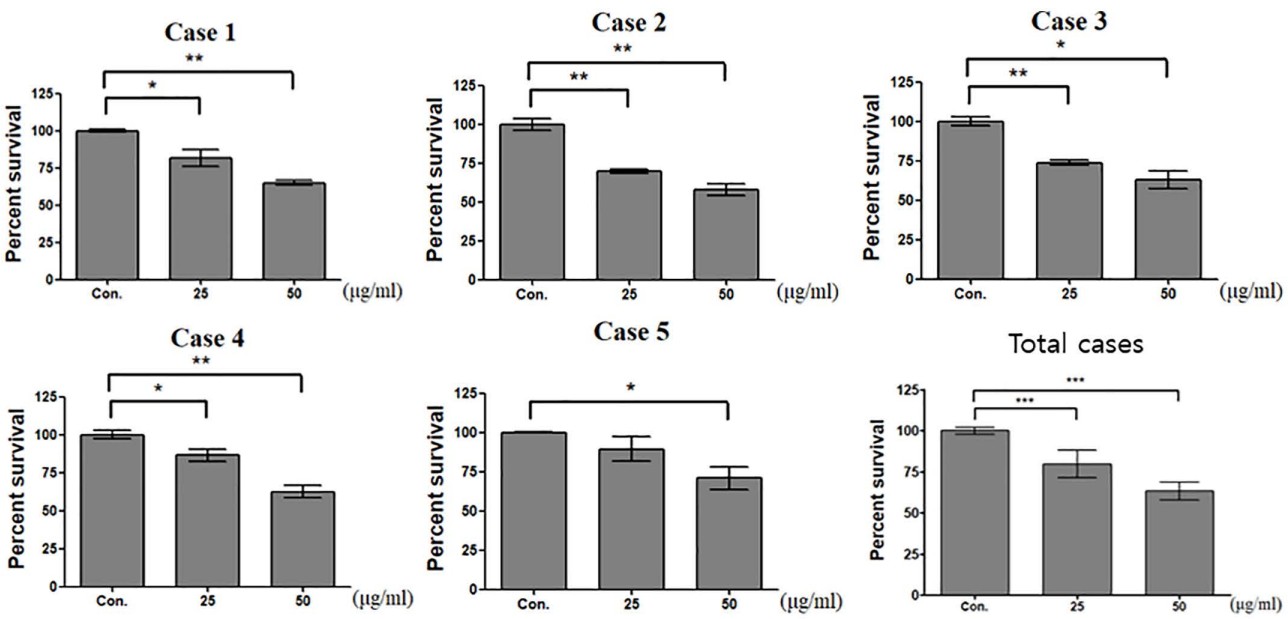

**Fig 3. Determination of optimal exosome treatment concentration.** Cell viability was measured 48 h after exosome treatment (25 μg/mL and 50 μg/mL) and independently assessed in triplicate for each case. Data are expressed as mean±SD. (* P<0.05,** P<0.01, and ***<0.001).

miRNA sequencing are illustrated in Fig 5D. The top 20 miRNAs included miR-21 and miR-125b, which are known to be associated with cachexia and were selected for subsequent experiments.

## Transfecting HSkM cells with miRNA isolated from CAF-derived exosomes

Among the 370 known miRNAs identified, miR-125b and miR-21 were selected for further analysis owing to their association with cachexia. We treated HSkM cells with varying concentrations of these miRNA mimics and assessed their viability. We observed no significant differences in cell viability across the tested concentrations, suggesting that miRNA transfection did not adversely affect the viability of HSkM cells (Fig 6). Transfecting HSkM cells with miR-125b mimics resulted in morphological changes similar to those induced by CAF-derived exosomes (Fig 7A). At all concentrations (10, 20, and 50 nM) of miRNA mimics, myosin diameter was significantly reduced compared with that in the mock group (Fig 7B). Notably, we observed no morphological changes, including reductions in myosin fiber thickness, in miR-21-transfected HSkM cells compared to mock-transfected controls (Fig 8A, 8B).

## Co-transfection of CAF-derived exosomes and miRNA inhibitor in HSkM cells

Treatment with the miR-125b mimic alone resulted in a marked reduction in myosin diameter and fiber thinning. However, co-treatment with miR-125b mimics and inhibitors induced no significant reduction in myosin diameter in HSkM cells, as evidenced by confocal microscopy (Fig 9A, 9B). We also co-treated HSkM cells with CAF-derived exosomes and a miR-125b inhibitor. Confocal microscopy revealed that treatment with exosomes alone resulted in a significant reduction in myosin diameter compared with that in the control. However, co-treatment with exosomes and miRNA inhibitors resulted in myosin diameters comparable to those of the mock control (Fig 10A). These findings, supported by statistical analysis, suggest that miR-125b from CAF-derived exosomes may contribute to a reduction in myosin diameter (Fig 10B).

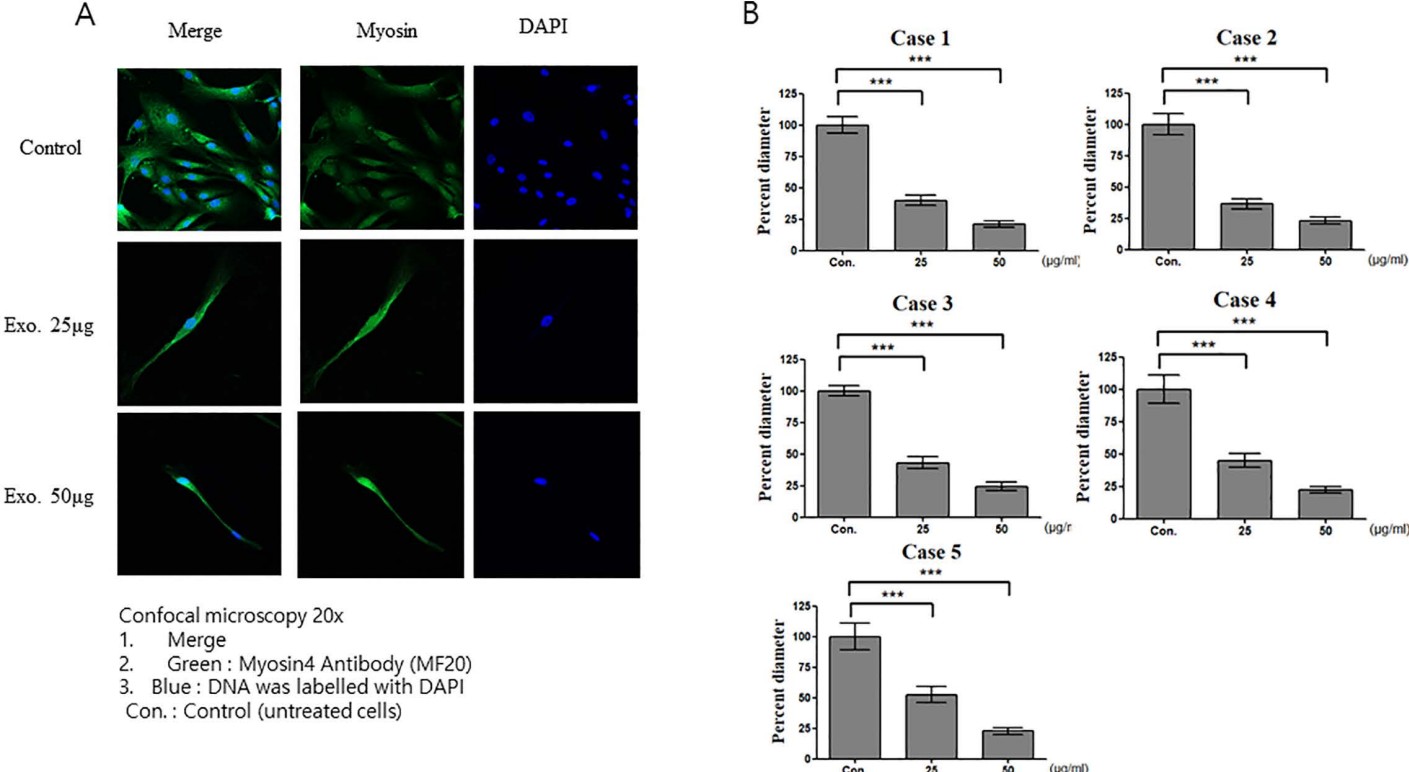

**Fig 4. Effects of CAF-derived exosomes on HSkM cells. (A)** Confocal microscopy images of untreated HSkM cells (control) or HSkM cells treated with 25 µg/mL and 50 µg/mL exosomes. The cells were stained for myosin (green) and nuclei (DAPI, blue). **(B)** Myosin diameter significantly decreased compared with that in the control group (p<0.001). Data are expressed as mean±SEM (n=15 measurements from one image). CAFs, cancer-associated fibroblasts; HSkM, human skeletal myoblasts.

## Discussion

Cancer cachexia is a multifactorial, irreversible syndrome characterized by progressive skeletal muscle wasting, weight loss, and multi-organ dysfunction. Various experimental models using skeletal muscle cell lines, such as HSkM, C2C12, and L6, have delineated key factors and mechanisms contributing to cachexia. These include tumor-derived factors, inflammatory cytokines (e.g., TNF-α, IL-6, and IL-8), hormonal imbalance (e.g., glucocorticoids, angiotensin, and myostatin), chemotherapeutic agents (e.g., cisplatin and doxorubicin), and miRNA-mediated gene regulation. These mediators promote catabolic signaling pathways, including the ubiquitin-proteasome system, autophagy, nuclear factor kappa-light-chain-enhancer of activated B cells (NF-κB), p38 mitogen-activated protein kinase (MAPK), and signal transducer and activator of transcription 3 (STAT3), resulting in enhanced protein degradation and impaired muscle regeneration [5–10]. However, previous studies have primarily focused on exosomes derived from tumor cells or circulating biofluids, with limited investigation into the effects of CAF-derived exosomes on skeletal muscle. The present study provides compelling in vitro evidence that exosomes derived from CAFs might induce structural changes in skeletal muscle cells, particularly by reducing the diameter of myosin fibers. Among the miRNAs enriched in CAF-derived exosomes, we identified miR-125b as a key molecule mediating this effect. Notably, the atrophic changes in myosin structure induced by exosomes were reversed upon co-treatment with a miR-125b inhibitor, suggesting a possible role of this miRNA in muscle wasting. These findings are consistent with recent insights indicating that CAFs, as major stromal components of the TME, may contribute to tumor progression and metastasis, as well as systemic metabolic syndromes such as cancer cachexia [11,24,25]. In

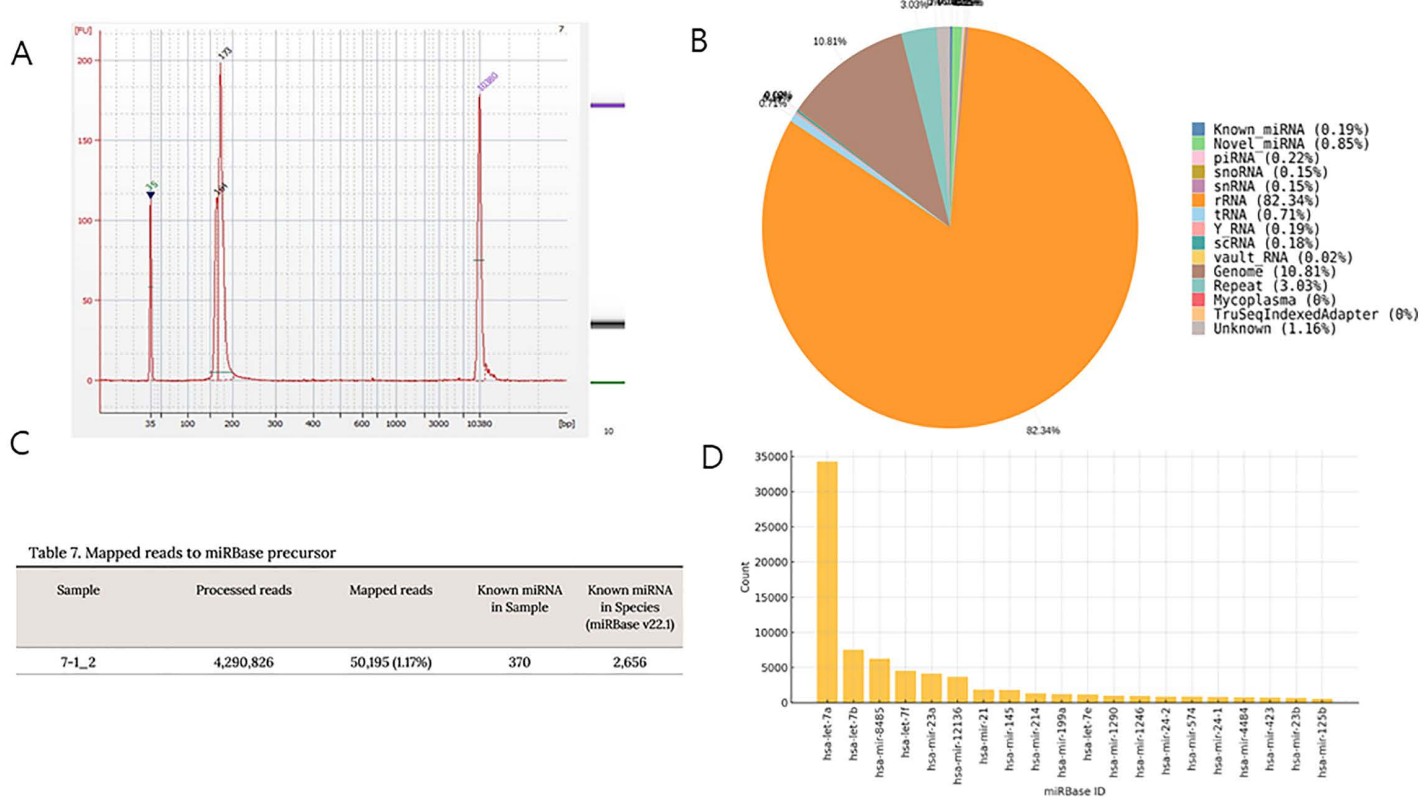

**Fig 5. miRNA analysis of CAF-derived exosomes from a single sample. (A)** Analysis with a high-sensitivity DNA chip run on an Agilent 2100 bioanalyzer demonstrates the quality and quantity of the gel-recovered library. **(B)** Small RNA composition of the sample. **(C)** Known miRNAs in the sample. **(D)** Expression levels of the top 20 miRNAs identified in CAF-derived exosomes. CAF, cancer-associated fibroblast; miRNA, microRNA.

this context, our findings extend the current understanding of the pathophysiology of cancer cachexia by demonstrating a link between CAF-derived exosomal miRNAs and myogenic atrophy. Our results point to the critical role of stromal cells within the TME as active contributors to systemic disease states and highlight their potential as therapeutic targets.

In this study, we investigated the effects of exosomes derived from CAFs on HSkM cells. Treatment with CAF-derived exosomes induced apoptosis in approximately 10–15% of these cells and led to a significant reduction in cell diameter, confirming the induction of cachexia. To elucidate the underlying mechanism, we sequenced the exosomal miRNAs and identified miR-125b as a candidate molecule potentially associated with cachexia. Treatment with miR-125b resulted in negligible apoptosis (less than 1%) in HSkM cells. However, we observed a notable decrease in cell diameter, which was more pronounced than that caused by the CAF-derived exosomes, indicating muscle atrophy. These findings suggest the following two points: First, miR-125b may be a key factor that induces muscle atrophy without inducing cell death. Second, the apoptosis observed following treatment with CAF-derived exosomes appears to be mediated by other bioactive molecules within the exosomes, rather than by miR-125b alone. Therefore, cachexia induced by CAF exosomes cannot be solely explained by the limited extent of apoptosis, suggesting that atrophy is a primary effect and distinct from general cytotoxicity.

The role of miRNAs in mediating cachexia-related muscle atrophy has garnered increasing attention. Several miRNAs are implicated in muscle and adipose tissue loss. For example, exosomal miR-26a can attenuate muscle atrophy

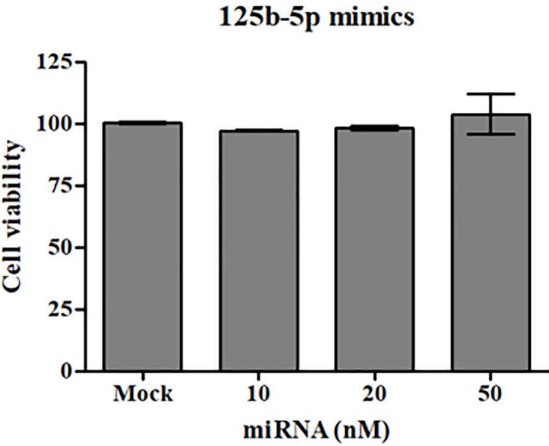

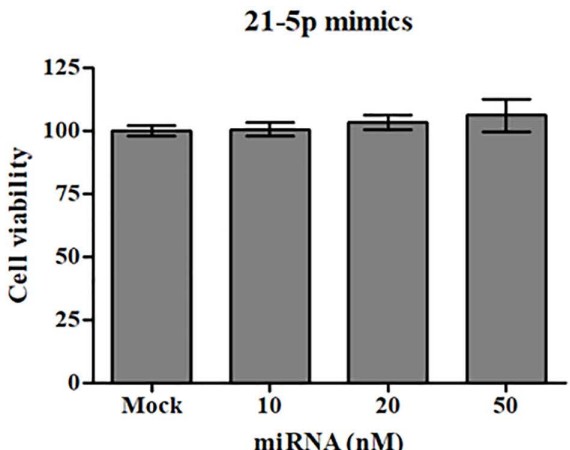

**Fig 6. Cell viability following treatment with miRNA mimics.** HSkM cells were treated with candidate miRNAs (miR-125b-5p and miR-21-5p), identified from CAF-derived exosomes, at concentrations of 10, 20, and 50 nM. Cell viability was assessed 48 h after treatment. No significant differences in cell viability were detected across the tested concentrations. Data are expressed as mean ± SD. CAF, cancer-associated fibroblast; HSkM, human skeletal myoblasts; miRNA, microRNA.

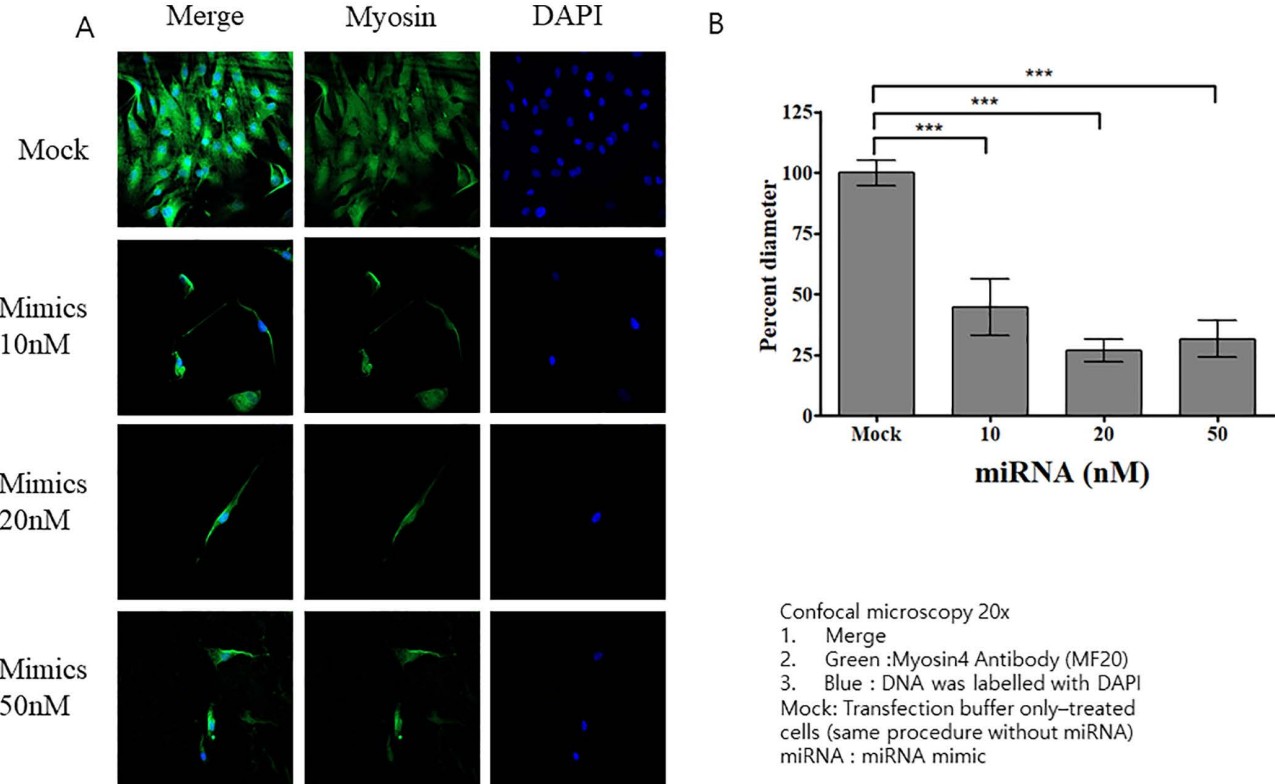

**Fig 7. Effects of the miR-125b mimic on HSkM cells. (A)** Confocal microscopy images of HSkM cells transfected with miR-125b or with a mock control. **(B)** At all concentrations of miR-125b mimic, myosin diameter was significantly reduced compared with that of the mock control-treated group (p < 0.001). Data are expressed as mean ± SEM (n = 15 measurements from one image). HSkM, human skeletal myoblasts.

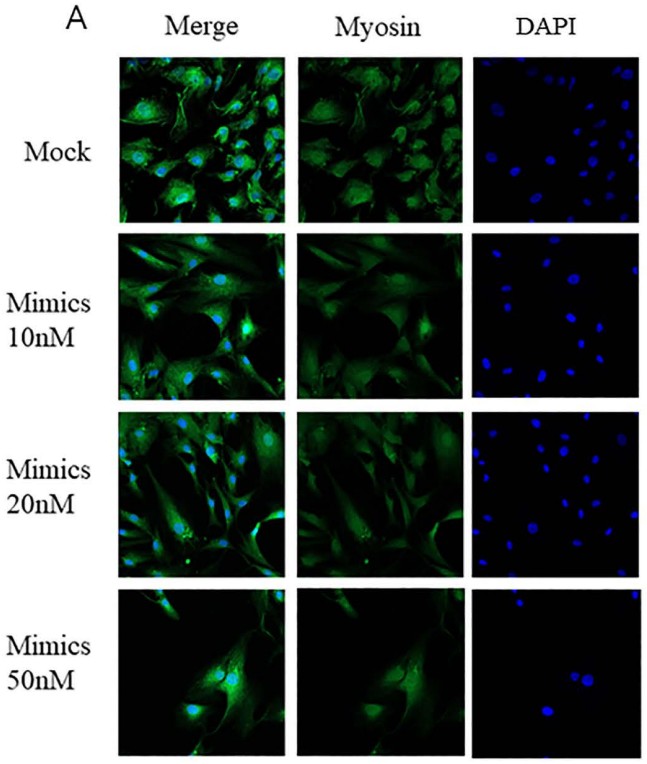
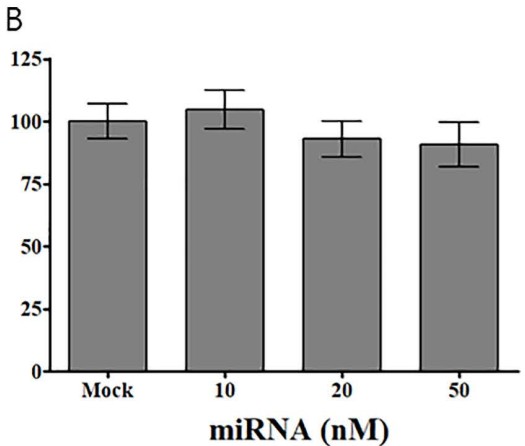

Confocal microscopy 20x
1. Merge
2. Green :Myosin4 Antibody (MF20)
3. Blue : DNA was labelled with DAPI
Mock: Transfection buffer only–treated cells (same procedure without miRNA)
miRNA : miRNA mimic

**Fig 8. Effects of the miR-21 mimic on HSkM cells. (A)** Confocal microscopy images of HSkM cells stained for myosin (green) and nuclei (DAPI, blue). Cells were transfected with miR-21 mimics at concentrations of 10, 20, and 50 nM for 48 **h. (B)** Quantification of myosin diameter. No morphological changes were observed for miR-21-transfected HSkM cells. Data are expressed as mean±SEM (n=15 measurements from one image). HSkM, human skeletal myoblasts.

by suppressing forkhead box protein O1 activity [26]. Moreover, miR-21, which is abundant in microvesicles derived from lung and pancreatic cancers, induces myoblast apoptosis by activating Toll-like receptor 7 (TLR7) and c-Jun N-terminal kinase signaling [27]. Exosomes from colorectal cancer cells enriched in miR-195a-5p and miR-125b-1-3p activate Bcl-2-mediated apoptosis in C2C12 myotubes [10]. Similarly, miR-29b, consistently overexpressed in atrophic muscle, impairs anabolic signaling by targeting insulin-like growth factor 1 and phosphoinositide 3-kinase [28]. Collectively, these studies support the role of miRNAs not just as biomarkers but as functional regulators in the progression of cancer cachexia. In our study, we selected miR-21 and miR-125b based on miRNA sequencing data and their reported association with cachexia. Contrary to our expectations, only miR-125b induced significant reductions in myosin diameter, whereas miR-21 did not elicit appreciable changes in muscle morphology or structure. This discrepancy may be attributed to tumor-type specificity or cell-of-origin differences. Specifically, most prior studies investigating miR-21 involved exosomes derived directly from lung and pancreatic tumor cells, whereas our study focused on CAF-derived exosomes from colon cancer tissues. These findings suggest that miRNA function may be context-dependent and modulated by the cellular origin and TME. Such observations warrant further investigation into the differential roles of miRNA-mediated signaling across various tumor types and stromal cell populations. Furthermore, the biological function and regulatory context of CAF-derived exosomal miR-125b may differ from those of tumor cell-derived or serum exosomal miRNAs. However, studies directly comparing these sources are extremely rare. Further research is required to clarify whether stromal-derived miRNAs exert cachexia-driving effects distinct from those of tumor-derived miRNAs or circulating vesicles.

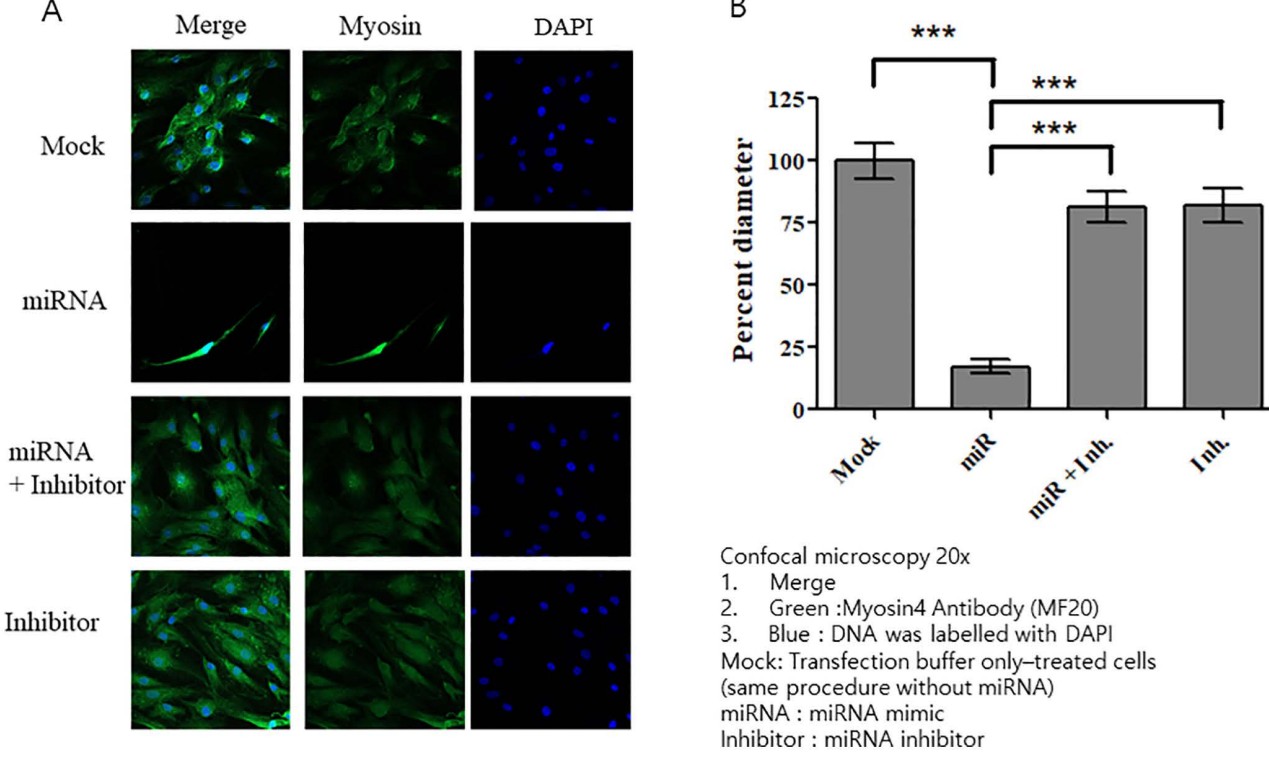

**Fig 9. Effects of combined treatment with the miR-125b mimic and inhibitor on HSkM cells. (A)** Confocal microscopy images of HSkM cells stained for myosin (green) and nuclei (DAPI, blue). Cells were treated with miR-125b mimic alone, miR-125b mimic plus inhibitor, or inhibitor alone for 48 **h. (B)** Treatment with miR-125b mimic alone resulted in a marked reduction in myosin diameter compared with that in the mock control (p < 0.001). Co-treatment with miR-125b mimics and inhibitors induced no significant reduction in myosin diameter. Data are expressed as mean ± SEM (n = 15 measurements from one image). HSkM, human skeletal myoblasts.

Exosomes serve as vehicles for miRNA transfer and may also independently contribute to muscle atrophy through other pathways [29]. Exosomes can activate inflammatory cascades in muscle tissue, including the upregulation of TNF-α and IL-6, which in turn activate NF-κB and STAT3 signaling [30]. These pathways disrupt cellular homeostasis, promote catabolism, and suppress muscle regeneration [31,32]. Moreover, exosomes can alter the function of other cells within the muscle microenvironment, such as fibroblasts and adipocytes, resulting in structural remodeling and metabolic dysfunction that collectively exacerbate muscle wasting [32]. This evidence is, to some extent, consistent with our findings. In our study, myosin preservation was more prominent in cells treated with miR-125b mimic and inhibitor than in those treated with exosomes and the same inhibitor, suggesting that other exosomal constituents may contribute to atrophy. Although our data indirectly support this hypothesis, further studies involving detailed profiling of exosomal components are necessary to validate this mechanism.

This study has a few limitations. First, the use of CAF samples from a single institution may introduce selection or reporting bias, and the relatively small sample size limits generalizability. Future studies involving larger stromal sample sets are necessary to validate reproducibility. Second, the interpretation of muscle atrophy was primarily based on morphological thinning of myosin fibers, and functional biomarkers of proteolysis such as atrogin-1, muscle RING-finger protein-1, myoblast determination protein 1, or myogenin were not quantified. Therefore, we cannot fully exclude the possibility that the observed phenotype reflects cytoskeletal remodeling rather than definitive cachexia-associated proteolytic activation. Third, downstream mechanistic pathways regulated by miR-125b were not characterized, and signaling

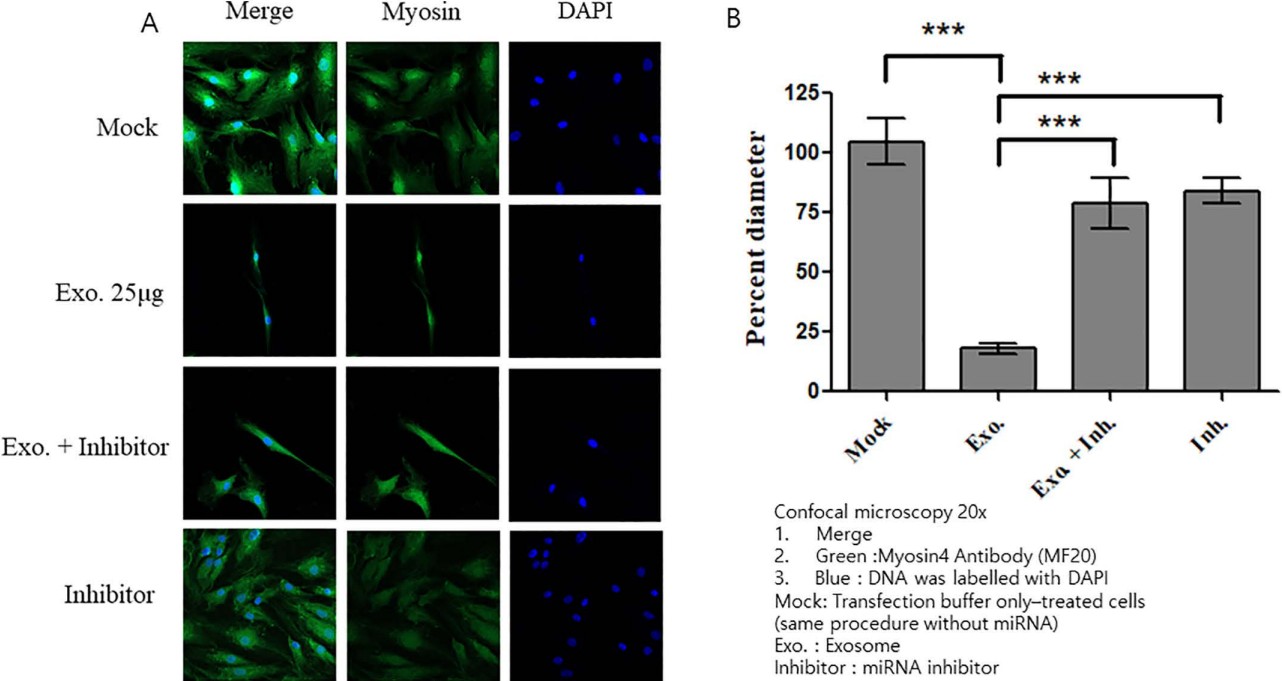

**Fig 10. Effects of CAF-derived exosomes and miR-125b inhibitor on HSkM cells. (A)** Confocal microscopy images of HSkM cells treated with mock control, CAF-derived exosomes (25 μg/mL), exosomes with miR-125b inhibitor, or miR-125b inhibitor alone. Cells were stained for myosin (green) and nuclei (DAPI, blue). **(B)** Treatment with exosomes alone resulted in a significant reduction in myosin diameter compared with that in the mock control (p < 0.001). Co-treatment with exosomes and miRNA inhibitors resulted in myosin diameters comparable to those of the mock control. Data are expressed as mean ± SEM (n = 15 measurements from one image). HSkM, human skeletal myoblasts.

interactions between CAF-derived vesicles and muscle cell proteostasis remain undefined. Fourth, miRNA sequencing was performed on a single sample, which may restrict the robustness of the miRNA profile. Moreover, we did not evaluate in vivo muscle wasting or systemic metabolic consequences; thus, the biological relevance of CAF-derived exosomal miR-125b requires further confirmation in animal models. Despite these limitations, our results provide novel evidence that CAF-derived exosomal miR-125b may contribute to skeletal muscle atrophy and suggest a stromal miRNA axis in cancer cachexia that has not been previously clarified.

To the best of our knowledge, this study is the first to experimentally demonstrate the role of CAF-derived exosomes in muscle atrophy. As research in oncology increasingly emphasizes the importance of the TME, a similar shift in cachexia research is warranted. Future studies incorporating various cell lines and in vivo models are required to validate our findings and assess systemic metabolic effects. As therapeutic strategies evolve beyond inflammation and metabolism, targeting intercellular communication through miRNAs and exosomes may represent a promising new avenue in the management of cancer cachexia.

## Supporting information

**S1 File. Raw data for figures.**
(XLSX)

**S2 Fig. Original Images for Blots.**
(PDF)

## Author contributions

**Conceptualization:** Bo-Young Oh, Kyung-Ah Cho, Soon Sup Chung, Gyoung Tae Noh.

**Data curation:** Ho Seung Kim, Jinsu Kim.

**Formal analysis:** Ho Seung Kim, Jinsu Kim, Bom Lee, Yonghyun Lee, So-Yeon Park.

**Funding acquisition:** Gyoung Tae Noh.

**Investigation:** Bo-Young Oh, Kyung-Ah Cho, Soon Sup Chung, Gyoung Tae Noh.

**Methodology:** Ho Seung Kim, Jinsu Kim, Gyoung Tae Noh.

**Project administration:** Gyoung Tae Noh.

**Resources:** Ho Seung Kim.

**Software:** Bom Lee, Yonghyun Lee, So-Yeon Park.

**Supervision:** Soon Sup Chung, Gyoung Tae Noh.

**Validation:** Jinsu Kim.

**Visualization:** Jinsu Kim, Bom Lee, Yonghyun Lee, So-Yeon Park.

**Writing – original draft:** Ho Seung Kim, Jinsu Kim.

**Writing – review & editing:** Gyoung Tae Noh.

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
