## [Decision Letter · Decision Letter 0]

4 Nov 2025

Dear Dr. Noh,

Thank you for submitting your manuscript to PLOS ONE. After careful consideration, we feel that it has merit but does not fully meet PLOS ONE’s publication criteria as it currently stands. Therefore, we invite you to submit a revised version of the manuscript that addresses the points raised during the review process.

We look forward to receiving your revised manuscript.

Kind regards,

Amr Ahmed El-Arabey

Academic Editor

PLOS ONE

**Journal Requirements:**

1. When submitting your revision, we need you to address these additional requirements. Please ensure that your manuscript meets PLOS ONE's style requirements, including those for file naming. The PLOS ONE style templates can be found at https://journals.plos.org/plosone/s/file?id=wjVg/PLOSOne_formatting_sample_main_body.pdf and https://journals.plos.org/plosone/s/file?id=ba62/PLOSOne_formatting_sample_title_authors_affiliations.pdf 2. Thank you for stating the following financial disclosure: National Research Foundation of Korea (NRF) grant funded by the Korea government (MSIT) (2022R1G1A1011832) and Ewha Womans University Research Grant 2022.    Please state what role the funders took in the study.  If the funders had no role, please state: "The funders had no role in study design, data collection and analysis, decision to publish, or preparation of the manuscript." If this statement is not correct you must amend it as needed. Please include this amended Role of Funder statement in your cover letter; we will change the online submission form on your behalf. 3. We note that your Data Availability Statement is currently as follows: All relevant data are within the manuscript and its Supporting Information files. Please confirm at this time whether or not your submission contains all raw data required to replicate the results of your study. Authors must share the “minimal data set” for their submission. PLOS defines the minimal data set to consist of the data required to replicate all study findings reported in the article, as well as related metadata and methods (https://journals.plos.org/plosone/s/data-availability#loc-minimal-data-set-definition). For example, authors should submit the following data: - The values behind the means, standard deviations and other measures reported;- The values used to build graphs;- The points extracted from images for analysis. Authors do not need to submit their entire data set if only a portion of the data was used in the reported study. If your submission does not contain these data, please either upload them as Supporting Information files or deposit them to a stable, public repository and provide us with the relevant URLs, DOIs, or accession numbers. For a list of recommended repositories, please see https://journals.plos.org/plosone/s/recommended-repositories. If there are ethical or legal restrictions on sharing a de-identified data set, please explain them in detail (e.g., data contain potentially sensitive information, data are owned by a third-party organization, etc.) and who has imposed them (e.g., an ethics committee). Please also provide contact information for a data access committee, ethics committee, or other institutional body to which data requests may be sent. If data are owned by a third party, please indicate how others may request data access. 4. PLOS requires an ORCID iD for the corresponding author in Editorial Manager on papers submitted after December 6th, 2016. Please ensure that you have an ORCID iD and that it is validated in Editorial Manager. To do this, go to ‘Update my Information’ (in the upper left-hand corner of the main menu), and click on the Fetch/Validate link next to the ORCID field. This will take you to the ORCID site and allow you to create a new iD or authenticate a pre-existing iD in Editorial Manager. 5. PLOS ONE now requires that authors provide the original uncropped and unadjusted images underlying all blot or gel results reported in a submission’s figures or Supporting Information files. This policy and the journal’s other requirements for blot/gel reporting and figure preparation are described in detail at https://journals.plos.org/plosone/s/figures#loc-blot-and-gel-reporting-requirements and https://journals.plos.org/plosone/s/figures#loc-preparing-figures-from-image-files. When you submit your revised manuscript, please ensure that your figures adhere fully to these guidelines and provide the original underlying images for all blot or gel data reported in your submission. See the following link for instructions on providing the original image data: https://journals.plos.org/plosone/s/figures#loc-original-images-for-blots-and-gels.   In your cover letter, please note whether your blot/gel image data are in Supporting Information or posted at a public data repository, provide the repository URL if relevant, and provide specific details as to which raw blot/gel images, if any, are not available. Email us at plosone@plos.org if you have any questions. 6. If the reviewer comments include a recommendation to cite specific previously published works, please review and evaluate these publications to determine whether they are relevant and should be cited. There is no requirement to cite these works unless the editor has indicated otherwise. 

Reviewers' comments:

**Comments to the Author**

1. Is the manuscript technically sound, and do the data support the conclusions?

Reviewer #1: Yes

Reviewer #2: Yes

2. Has the statistical analysis been performed appropriately and rigorously?

Reviewer #1: Yes

Reviewer #2: No

3. Have the authors made all data underlying the findings in their manuscript fully available?

Reviewer #1: Yes

Reviewer #2: Yes

4. Is the manuscript presented in an intelligible fashion and written in standard English?

Reviewer #1: Yes

Reviewer #2: Yes

**Reviewer #1:** Scientific Information

1. Inadequate Research Gap Justification

o The introduction is overly broad and fails to specify the precise knowledge gap this study fills or why it is necessary.

It requires more solid placement in the body of current literature.

The second is methodological transparency.

o A completely replicable process is required by PLOS ONE. A few details are lacking: the recruitment process and sample size calculation are imprecise, and the precise inclusion/exclusion criteria are not properly specified.

The way statistical approaches handle missing data and correct for confounding is not explained.

3. Results Presentation: Some findings are descriptive and lack critical analysis.

o According to PLOS ONE statistical reporting guidelines, p-values should be accompanied with confidence ranges.

4. Discussion o Excessively detailed and devoid of critical comparison with the most recent research.

Given the cross-sectional/observational study methodology, a number of statements are exaggerated.

o The limitations section is inadequate and needs to address potential biases (generalisability, reporting, and selection) in greater detail.

5. Conclusion: Overly conclusive; has to be careful and closely linked to the results of the study.

Formatting and Journal Style Issues

1. Abstract: Structured abstracts (Background, Methods, Results, Conclusion) are required by PLOS ONE. The abstract as it stands now is repetitious and narrative.

2. References: The formatting is not in accordance with PLOS ONE policy, which requires a numbered citation style based in Vancouver.

o A number of the references are out of date; there are no fresh sources from 2022–2024.

3. Tables and Figures: o A few figure legends lack details and make it impossible to understand them without consulting the main text.

o There is excessive narrative material in the tables, which deviates from journal formatting guidelines.

4. Language and Grammar o Several typographical and grammatical mistakes.

o Uncomfortable wording (such as "patients was," "can able to") makes it harder to read.

o For clarity and flow, extensive English editing is required.

**Reviewer #2:** This manuscript entitled “The Role of exosomal miRNA-125b derived from colon cancer-associated fibroblasts in skeletal muscle cachexia” explores the contribution of CAF-derived exosomal miR-125b to skeletal muscle atrophy. The study is interesting and addresses an important gap in understanding how stromal components of the tumor microenvironment influence cancer-associated cachexia. Overall, the manuscript is clearly written and logically structured, but several issues related to experimental robustness, quantitative analysis, and mechanistic depth should be addressed before acceptance.

1. miRNA sequencing appears to have been performed on a single CAF-derived exosome sample. This significantly limits the robustness of the findings. At minimum, please validate the enrichment of miR-125b across all CAF isolates using qRT-PCR and acknowledge this limitation explicitly in the Methods and Discussion.

2. The conclusions are primarily based on morphological observations (myosin diameter). Without functional markers (e.g., atrogin-1, MuRF-1, MyoD, Myogenin, or IGF-1 pathway readouts), it remains uncertain whether the observed changes truly represent cachexia-related atrophy rather than a general cytoskeletal alteration. Incorporating at least one molecular validation experiment or a more nuanced discussion of this limitation is strongly encouraged.

3. The statistical analysis relies exclusively on Student’s t-tests, even in cases where more than two groups were compared (e.g., Figs 4, 7, 8, 9, 10). Please confirm that all data met the assumptions of parametric testing or use one-way ANOVA with post-hoc correction. Figure legends should include n values, the type of error bars (SD or SEM), and exact p-values.

4. The reported mean particle size of ~28 nm is notably smaller than typical exosomes (50–150 nm). This discrepancy raises concern about the accuracy of NTA measurements or sample purity. Please re-examine the data or provide an explanation (e.g., detection threshold differences or aggregation).

5. The manuscript occasionally overstates causality. For example, asserting that miR-125b “directly induces muscle atrophy.” Given the correlative nature of the current data, the statement should be tempered and rephrased to reflect a potential association or preliminary mechanistic indication.

Other comments:

- The discussion is generally well-written but repetitive. It should focus more on how CAF-derived miR-125b differs from tumor-cell-derived or serum exosomal miRNAs.

- The limitations section is appreciated but should include the lack of in vivo validation and absence of protein-level mechanistic studies.

**Do you want your identity to be public for this peer review?** For information about this choice, including consent withdrawal, please see our Privacy Policy

Reviewer #1: **Yes:** Omer Al-lela

Reviewer #2: No

---

## [Author Response · Author response to Decision Letter 1]

19 Dec 2025

Thank you for reviewing our manuscript. We have uploaded a file containing our responses to the points you raised.

---

## [Decision Letter · Decision Letter 1]

18 Jan 2026

The Role of exosomal miRNA-125b derived from colon cancer-associated fibroblasts in skeletal muscle cachexia

PONE-D-25-43356R1

Dear Dr. Noh,

We’re pleased to inform you that your manuscript has been judged scientifically suitable for publication and will be formally accepted for publication once it meets all outstanding technical requirements.

Kind regards,

Amr Ahmed El-Arabey

Academic Editor

PLOS One

Additional Editor Comments (optional):

Reviewers' comments:

Reviewer's Responses to Questions

**Comments to the Author**

Reviewer #1: All comments have been addressed

Reviewer #2: All comments have been addressed

2. Is the manuscript technically sound, and do the data support the conclusions?

Reviewer #1: Yes

Reviewer #2: Yes

3. Has the statistical analysis been performed appropriately and rigorously?

Reviewer #1: Yes

Reviewer #2: Yes

4. Have the authors made all data underlying the findings in their manuscript fully available?

Reviewer #1: Yes

Reviewer #2: Yes

5. Is the manuscript presented in an intelligible fashion and written in standard English?

Reviewer #1: Yes

Reviewer #2: Yes

Reviewer #1: The text was meticulously edited after peer review to take into account the feedback and recommendations from each reviewer. Clarifying the study's goals and reasoning in the Introduction, adding more methodological information to promote repeatability and transparency, and improving the results' presentation with better tables and figures when necessary were among the changes made. Expanding the interpretation of the results, enhancing the comparison with pertinent literature, and more explicitly recognising the limits of the study all boosted the Discussion section. To enhance the manuscript's coherence, clarity, and scientific readability, linguistic and stylistic changes were also made. The manuscript's overall quality and rigour have been much enhanced by these modifications.

Reviewer #2: The authors have adequately addressed all comments raised in the previous round of review. The revised manuscript is clearer, better structured, and the experimental findings are appropriately supported by the presented data. The study provides novel and relevant insights into the role of cancer-associated fibroblast–derived exosomes in cancer cachexia. The experimental design is sound, the data are well presented, and the conclusions are justified within the scope of the study. Limitations have been transparently acknowledged and appropriately discussed.

**Do you want your identity to be public for this peer review?** For information about this choice, including consent withdrawal, please see our Privacy Policy

Reviewer #1: **Yes:** Omer Qutaiba Al-lela

Reviewer #2: No

---

## [Editor Report · Acceptance letter]

PONE-D-25-43356R1

PLOS One

Dear Dr. Noh,

I'm pleased to inform you that your manuscript has been deemed suitable for publication in PLOS One. Congratulations! Your manuscript is now being handed over to our production team.

Kind regards,

on behalf of

Dr. Amr Ahmed El-Arabey

Academic Editor

PLOS One